# The Preparation of Gen-NH_2_-MCM-41@SA Nanoparticles and Their Anti-Rotavirus Effects

**DOI:** 10.3390/pharmaceutics14071337

**Published:** 2022-06-24

**Authors:** Lijun Song, Jiabo Chen, Yuxuan Feng, Yujing Zhou, Feng Li, Guiqin Dai, Yue Yuan, Haosen Yi, Yupei Qian, Siyan Yang, Yang Chen, Wenchang Zhao

**Affiliations:** Guangdong Provincial Key Laboratory of Research and Development of Natural Drugs, School of Pharmacy, Guangdong Medical University, Dongguan 523808, China; slj@gdmu.edu.cn (L.S.); m18630572001@163.com (J.C.); fyx199810@163.com (Y.F.); zyj_yjs927@163.com (Y.Z.); lffootball@163.com (F.L.); daiguiqin0518@163.com (G.D.); z97883481@163.com (Y.Y.); yikhosam@163.com (H.Y.); qyp0032@126.com (Y.Q.); siyany969@163.com (S.Y.); ychan227@163.com (Y.C.)

**Keywords:** genistein, mesoporous silica, Caco-2 cells, sodium alginate, targeted release

## Abstract

Genistein (Gen), a kind of natural isoflavone drug monomer with poor water solubility and low oral absorption, was incorporated into oral nanoparticles with a new mesoporous carrier material, NH_2_-MCM-41, which was synthesized by copolycondensation. When the ratio of Gen to NH_2_-MCM-41 was 1:0.5, the maximum adsorption capacity of Gen was 13.15%, the maximum drug loading was 12.65%, and the particle size of the whole core–shell structure was in the range of 370 nm–390 nm. The particles were characterized by a Malvern particle size scanning machine, XRD, Fourier transform infrared spectroscopy, scanning electron microscopy, and nitrogen adsorption and desorption. Finally, Gen-NH_2_-MCM-41 was encapsulated by sodium alginate (SA), and the chimerism of this material, denoted as GEN-NH_2_-MCM-41@SA, was investigated. In vitro release experiments showed that, after 5 h in artificial colon fluid (pH = 8.0), the cumulative release reached 99.56%. In addition, its anti-rotavirus (RV) effect showed that the maximum inhibition rate was 62.24% at a concentration of 30 μM in RV-infected Caco-2 cells, and it significantly reduced the diarrhea rate and diarrhea index in an RV-infected-neonatal mice model at a dose of 0.3 mg/g, which was better than the results of Gen. Ultimately, Gen-NH_2_-MCM-41@SA was successfully prepared, which solves the problems of low solubility and poor absorption and provides an experimental basis for the application of Gen in the clinical treatment of RV infection.

## 1. Introduction

Gen (4,5,7-trihydroxy-isoflavone), also called dye lignin (its chemical structure is shown in Figure 1), which has been found in several plants, including lupin, fava beans, soybeans, and so on, has shown multiple pharmacological effects on tumors, cardiovascular diseases, diabetes, and osteoporosis [1,2,3], and it has also had good activity on a variety of viruses, such as herpes simplex virus, bovine viral diarrhea virus, simian virus, and Epstein–Barr virus [4,5]. In recent years, our research group found that Gen could significantly inhibit the RNA transcription and viral protein synthesis of RV in RV-infected Caco-2 cells, which is mainly regulated by the cAMP/PKA/CREB signaling pathway [6,7]. However, its low solubility in water and lability toward acid degradation when taken via the oral route seriously hinders its clinical application and potential druggability.

MCM-41 (Mobil Composition of Matter No. 41) is a mesoporous material with a hierarchical structure(as shown in Figure 2)from a family of silicate and aluminosilicate solids that were first developed by researchers at Mobil Oil Corporation and can be used as catalysts or catalyst supports [8].

MCM-41 consists of regular cylindrical mesopores that form a one-dimensional pore system characterized by an independently adjustable pore diameter, sharp pore distribution, large surface, and large pore volume [9]. The pore size is larger than that of zeolite, and the pore distribution makes it easy to adjust the mesoporous diameter of 2~6.5 nm [10].

As we know, nanodrug delivery systems have attracted more and more attention because of their ability to change drug absorption, distribution, and metabolism in vivo. The mesoporous silica nanoparticle MCM-41 is one of the emerging drug-loading materials in recent years, with the advantages of a large specific surface area, regular pore structure, adjustable pore size, and large capacity [11]. By modifying the surface structure of mesoporous silica (as shown in Figure 3), a targeted drug release effect can be achieved, and the drug can be delivered accurately to the lesion site, with little or even zero leakage in normal tissues, thereby reducing its toxicity and adverse reactions. In our experiment, amino modification of MCM-41 was employed via copolycondensation. During the synthesis of mesoporous silicon particles, TEOS was directly added with a silane coupling agent, and hydrolytic polycondensation occurred under the protection of the template. Pore entrance modification was mainly carried out based on the dense organic layer at the entrance of MCM-41 after the quick modification of organic functional groups. This method is also convenient for the subsequent realization of amino-modified MCM-41 through the protonation of amino groups under different pH conditions to achieve the purpose of drug release.

The design starting point of pH-responsive MCM-41 is the use of the pH difference in the gastrointestinal tract [12]. For the drug release principle of pH-responsive mesoporous silica, the main research direction is to focus on changes in the organic chain under acidic conditions, which is mainly divided into two types. One is the gradual increase in H^+^ concentration under acidic conditions, which destroys the force between organic chains on the surface of MCM-41 [13]. After the chain breaks, the pore openings open, and the drug leaks out. The other is the change in the attraction between the functional groups of the chain itself under acidic conditions; the chain shrinks, the pore crossing is exposed, and the drug can be released [14].

Nevertheless, the encapsulant for porous mesoporous silicon drug loading is necessary for reducing drug leakage, and SA, as the encapsulant, has the advantages of biocompatibility and charged polymer. Importantly, SA is slightly soluble in aqueous solution, and the pH of 1% SA in aqueous solution is about 7.2 [15], which matches well with the pH value of the human body. The pH sensitivity of SA is due to the presence of carboxyl groups on its surface [16]. At low pH, the ionization degree of carboxyl groups decreases, resulting in increased hydrophilicity, SA shrinks to form an insoluble film, and the encapsulated drug cannot be released; in the intestinal environment with nearly weak alkaline pH, the dense film disintegrates rapidly, and the drug can be released [17]. Furthermore, SA and NH_2_-MCM-41 can be closely combined with van der Waals forces through hydrogen bonds, making them less soluble or insoluble in the stomach, so they release the drug in the intestine.

In this project (as shown in Figure 4), NH_2_-MCM-41&SA was employed to prepare inclusion complexes with Gen based on the principle of valence bond adsorption between the NH_2_ group and COOH on the SA surface [18]. In addition, SA further strengthened the pH response of whole nanoparticles, finally forming Gen-NH_2_-MCM-41 as the core and SA as the shell in a core–shell structure. This structure not only had good stability but also could effectively prevent agglomeration between nanoparticles so as to achieve the effect of complementary advantages between the core and shell. Accordingly, the pH-dependent drug release and the kinetics of drug release from NH_2_-MCM-41&SA were estimated in vitro. Its anti-RV effect was evaluated in RV-infected Caco-2 cells and a neonatal mouse model in vitro and in vivo.

## 2. Materials and Methods

### 2.1. Materials

#### 2.1.1. Chemical Experimental Materials

Methanol, ethanol, ethyl acetate, sodium hydroxide, hydrochloric acid, N, N-dimethylformamide, and dimethyl sulfoxide were purchased from Tianjin Damao Chemical Reagent Factory (Tianjin, China). Ethyl orthosilicate, cetyl trimethyl ammonium bromide, 3-aminopropyltriethoxysilane, ammonia water, SA, sodium dihydrogen phosphate, and sodium monohydrogen phosphate were purchased from Aladdin Reagent Co., Ltd. (Shanghai, China). Gen was purchased from Shanghai Macleans Biochemical Technology Co., Ltd. (Shanghai, China).

#### 2.1.2. Biological Cell Experimental Materials

Fetal bovine serum (Hyclone) was purchased from Thermo Scientific (Waltham, MA, USA); high-sugar DMEM medium was purchased from GIBCO (Waltham, MA, USA); penicillin and streptomycin, dimethyl sulfoxide, MTT reagent, and 0.25% trypsin digest were purchased from Solarbio company (Beijing, China); and 1× phosphate buffer (PBS) was purchased from Jiangsu biyuntian Biotechnology Research Institute (Jiangsu, China). Finally, 6 mm cell culture dishes and 6-, 12-, 24-, and 96-hole plates were purchased from Corning-Coster Company (Corning, NY, USA).

#### 2.1.3. Animal Experimental Materials

HE staining kit, 4% paraformaldehyde, and neutral resin were obtained from Beijing Solarbio Technology Co., Ltd. (Beijing, China). Hydrogen peroxide solution, absolute ethyl alcohol, and xylene were purchased from Tianjin Damao Chemical Reagent Factory (China, Tianjin). BSA was purchased from an American sigma company (St. Louis, MO, USA). The embedding agent 812 was purchased from the SPI company (China, Shanghai). DAPI, 2.5% glutaraldehyde, and acetone were purchased from Sinopharm Chemical Reagent Co., Ltd. (Beijing, China). Human Rotavirus Kit (ELISA) was purchased from Shanghai Zhuocai Biotechnology Co., Ltd. (Shanghai, China).

### 2.2. Synthesis

Gen-NH_2_-MCM-41 was prepared by the impregnation method and then encapsulated into core–shell structures by using the principle of valence bond adsorption between -COOH and -NH_2_ on the SA surface.

#### 2.2.1. Preparation of MCM-41

First, 1.0 g of CTAB was evenly dispersed in 100 mL of deionized water and 70 mL of ammonia and then heated to 60 °C in a water bath. After it was completely dissolved, 5 mL of TEOS was slowly added under magnetic stirring, and stirring continued for 7 h. Then, the reaction was terminated, and the solution was crystallized at room temperature for 7 h, centrifuged at 8000 rpm for 6 min, washed with 200 mL of absolute ethanol 3 times and 200 mL of deionized water 3 times, and freeze-dried. A 1.0 g dried sample was taken and added to 100 mL of a 2.5% by mass hydrochloric acid ethanol solution, which was heated to reflux for 24 h to remove the template, then washed with deionized water to neutrality, and vacuum-dried to obtain MCM-41.

#### 2.2.2. Preparation of NH_2_-MCM-41

A total of 0.28 g of NaOH was added to 480 mL of deionized water and mixed well, and then 1.0 g of CTAB was added under vigorous stirring and heated to 80 °C. After stirring for 30 min, 5.0 mL of TEOS was added dropwise and refluxed at 80 °C [19]. The reaction proceeded for 2 h. After the reaction, the product was centrifuged at 9500 rpm for 8 min to obtain a white product, which was then ultrasonically cleaned with methanol 3 times and then with deionized water 3 times. After freeze-drying, the mesoporous silicon nanoparticles with the surfactant template CTAB (CTAB@MCM) were obtained [12].

A total of 1.20 g of CTAB@MCM synthesized in the previous step was accurately weighed and evenly dispersed in 120 mL of anhydrous methanol. After adding 4.00 mL of APTES, the solution was stirred vigorously for 24 h and then centrifuged at 9500 rpm for 8 min [20]. Then, it was washed 3 times with 200 mL of anhydrous methanol and lyophilized to obtain CTAB@MCM-NH_2_-41. Using the methanol acid extraction method to remove the CTAB template: 1.0 g of CTAB@MCM-NH_2_-41 was accurately weighed, and 160 mL of anhydrous methanol was added and sonicated for 20 min to make the dispersion uniform; then, 9.0 mL of concentrated hydrochloric acid (Mw = 37.5%) was added, stirred vigorously overnight, heated to 60 °C, and then refluxed for 48 h. At the end, the mixture was centrifuged at 8000 rpm for 5 min, then washed with 200 mL of anhydrous methanol 3 times, and freeze-dried [21].

### 2.3. Sample Preparation and Analysis

#### 2.3.1. NH_2_-MCM-41 Loading with Gen

The adsorption method was used for drug loading. A certain amount of Gen was dissolved in ethyl acetate solvent and mixed with NH_2_-MCM-41 according to a ratio of drug to carrier of 1:0.5, 1:1, and 1:2, and magnetic stirring was used for 6 h to achieve adsorption. After equilibration, the mixture was centrifuged to remove uncoated Gen. The supernatant was removed, and the solvent was evaporated. The product was dried to obtain a powder under vacuum at 25 °C for 12 h, ground, and sieved to obtain the NH_2_-MCM-41-coated Gen drug carrier as the sample Gen-NH_2_-MCM-41. The maximum drug loading ratio was compared by calculation [1].

#### 2.3.2. SA Packaging of Gen-NH_2_-MCM-41

A total of 1 g of SA was dissolved in 100 mL of water and stirred until it was completely dissolved to prepare a 1% SA aqueous solution. A volume of 10 mL of SA aqueous solution was taken, put in 100 mg of Gen-NH_2_-MCM-41, ultrasonically dispersed for 4 h to reach adsorption equilibrium, and centrifuged. The supernatant was removed, and the obtained solid powder was placed at 25 °C, vacuum-dried for 12 h, ground, sieved, and vested.

#### 2.3.3. Gen-NH_2_-MCM-41@SA In Vitro Release Performance Test

To determine in vitro release, the basket method in the *Pharmacopoeia of the People’s Republic of China* (2020 edition, part IV dissolution and release determination) was used. A volume of 900 mL of synthetic gastric juice (pH = 1, 0.1 mol/L HCL), 900 mL of synthetic intestinal fluid (pH = 6.8 phosphate buffer), and 900 mL of synthetic colon fluid (pH = 7.4 phosphate buffer) were used as dissolution media. The experimental conditions were set as a rotating basket speed of 50 rpm and a temperature of 37 ± 0.5 °C. The experimental procedure is as follows: 5.00 g of Gen-NH_2_-MCM-41@SA was weighed in a rotating basket, and artificial gastric juice was added; a sample of 5 mL was taken after 1 h and 2 h, and a 0.22 μm filter was used to filter the sample. Subsequently, the synthetic gastric juice was carefully poured out and replaced with artificial intestinal juice. Samples of 5 mL were taken at 3 h, 4 h, and 5 h and passed through a 0.22 μm filter membrane. After each sample, an equal amount of isothermal dissolution medium was added. The artificial intestinal fluid was poured out after 5 h and replaced with synthetic colon fluid [2]. Samples of 5 mL were taken at 6 h, 8 h, 10 h, 12 h, and 14 h, and each sample was passed through a 0.22 μm filter membrane. After each sample, an equal amount of isothermal solution was added to dissolve the medium. Samples were taken to measure the ultraviolet absorption, and the results were averaged from 3 parallel experiments [22].

#### 2.3.4. Gen-NH_2_-MCM-41@SA Drug Release Model Fitting

The data release model was fitted for the release rate, and the release mechanism of Gen-NH_2_-MCM-41@SA was analyzed. The models used mainly include [23]:

Zero-order model:$$\frac{M_{t}}{M_{\infty }}=K_{0}t+Q_{0}$$

First-order model:$$\ln(1-\frac{M_{t}}{M_{\infty }})=K_{0}t$$

Higuchi model:$${K_{0}t}^{\frac{1}{2}}=\frac{M_{t}}{M_{\infty }}$$

Korsmeyer–Peppas kinetic model:$$\frac{M_{t}}{M_{\infty }}=K_{0}t^{n}$$

*K* is the fitting constant of each drug release model, $\frac{M_{t}}{M_{\infty }}$ expresses the cumulative drug release within *t*, Q_0_ is the initial concentration of the drug, *K*_0_ is the corresponding kinetic constant, and n is the release index.

### 2.4. Physico-Chemical Characterization

#### 2.4.1. XRD

Powder X-ray Diffraction (XRD) patterns were collected on an X’Pert Pro Bragg Brentano diffractometer (Philips, Milan, Italy). A copper target Ka ray was used, the working voltage was 40 kV, the tube current was set to 30 mA, the scanning range was set to 3–5°, and the scanning speed was set to 0.5°/min. Then, the sample was ground into the sample test piece and lightly pressed. After the surface was flat, it was placed in an X-ray diffractometer to characterize the crystal structure and crystal form of silica [24].

#### 2.4.2. Nitrogen Adsorption–Desorption (BET)

The synthesized sample was weighed and put into the sample tube. After the instrument was preheated, the sample was placed at 180 °C for degassing, and then a nitrogen adsorption and desorption experiment was conducted at 77 K to obtain the adsorption and desorption isotherm. The specific surface area was calculated by processing the BET adsorption isothermal equation. The mesoporous pore size distribution was calculated by the BJH method and the Barret, Joyner, and Halenda method. The pore size distribution and pore capacity were obtained by processing the data [25].

The BET equation is as follows:$$V=\frac{V_{m}CP}{\left(P_{0}-P)[1+(C-1)(P/P_{0})\right]}$$
where *P* is the equilibrium pressure of adsorbed gas; *P*_0_ is the saturated vapor pressure of adsorbed gas at the same temperature; *V* is the volume of adsorbed gas; *V*_m_ is the volume of adsorbate during saturated adsorption of the monolayer; and *C* is a constant (related to adsorption heat).

#### 2.4.3. Fourier Infrared Spectroscopy (FT-IR)

The measurement was performed using the KBr tablet method [26]. The prepared sample was ground with KBr according to a certain ratio and then compressed. The scanning wavelength was set to 4000–450 cm^−1^, and the spectrum was measured using MagnaIR500 II [4].

#### 2.4.4. Zeta Potential

The prepared sample was dispersed in a small amount of aqueous solution by ultrasound, and then the Zeta potential was measured with a Malvern laser particle size analyzer [27].

#### 2.4.5. Scanning Electron Microscopy (SEM)

A small amount of sample was taken and dispersed in absolute ethanol. After ultrasonic dispersion treatment, the treated sample was dropped onto a silicon wafer with a dropper and then placed at room temperature for drying. After spraying it with gold, the surface morphology of the sample was observed with a scanning electron microscope [28].

#### 2.4.6. Differential Scanning Thermal Analysis (DSC)

Gen, Gen-NH_2_-MCM-41, the Gen-NH_2_-MCM-41&SA physical mixture, and Gen-NH_2_-MCM-41@SA nanoparticles were taken separately and placed in a Pt crucible, and the nitrogen flow rate was set to 50 mL/min. The working voltage was 0.1 Mpa, and the temperature was increased from 0 °C to 180 °C at 5 °C/min [29]. The sample weighed no more than 5 mg. During the experiment, an empty Pt crucible plate was used as a blank control, and the absorption curve was recorded. After the experiment was over, the next set of experiments was carried out after cooling to room temperature [9].

#### 2.4.7. Adsorption Performance and Drug Loading Test (UV)

The prepared aminated mesoporous silica was analyzed by ultraviolet spectrophotometry to determine its adsorption capacity for Gen. The UV wavelength of Gen is 271 nm. The UV absorption standard curve of Gen methanol solution measured by the concentration gradient method is y = 12.571x + 0.004 (R^2^ = 0.9912). Then, the concentration change of Gen in the solution was measured by a UV spectrophotometer according to the following formula to calculate NH_2_-MCM-41 adsorption capacity (AC) and drug-loading capacity (LC) [30].
$$AC=\frac{m_{0}-\rho_{1}V_{1}}{m}$$
$$LC=\frac{m_{0}-\rho_{1}V_{1}}{m\times 1000+(m_{0}-\rho_{1}V_{1})}$$
where m_0_ is the mass of Gen added to the solution (mg); ρ_1_ is the mass concentration of Gen solution after carrier adsorption (mg/L); *V*_1_ is the volume of Gen solution after adsorption (*L*); and m is the mass (g) of NH_2_-MCM-41.

### 2.5. Administration of Gen-NH_2_-MCM-41@SA, NH_2_-MCM-41, SA, and Gen against RV Infection in Caco-2 Cells

#### 2.5.1. Cell Culture

Caco-2 and MA104 cells were cultured in a high-sugar DMEM medium containing 1% anti-penicillin, anti-streptomycin, and 10% FBS at 37 °C and 5% CO_2_. Cells were passaged when they reached 85–90% confluency [31].

#### 2.5.2. Amplification and Titer Determination of RV in MA104 Cells

RV was taken from −80 °C and thawed at 4 °C for virus amplification. Then, 10 g/mL trypsin without EDTA was added, incubated at 37 °C for 30 min, and then inoculated into MA104 cells. When the cytopathic effect (CPE) was about +++ [10,32], the cell culture flask was placed at −20 °C, and after 24 h, it was thawed at 4 °C. The above operations were repeated three times, after which the freeze–thawed cell solution was collected in a centrifuge tube and centrifuged at 4 °C and 12,000× *g* for 30 min. Finally, the harvested virus was collected and packaged in a sterile centrifuge tube, labeled, and stored at −80 °C [33].

For the titer determination of RV, the incubated RV was first diluted into a series of concentrations, including 10^−1^, 10^−2^, 10^−3^, 10^−4^, 10^−5^, and 10^−6^, in DMEM culture solution without fetal bovine serum and then applied to MA104 cells at 100 μL/well. The number of CPE-free wells was recorded for each dilution when a CPE was no longer present in the 96-well plates of RV with the lowest dilution. The median tissue culture infective dose (TCID_50_) of the virus was calculated using the Reed and Muench method [13].

#### 2.5.3. The Cytotoxic Effect of Gen-NH_2_-MCM-41@SA

The CCK-8 assay was used to detect the toxic effects of Gen-NH2-MCM-41@SA on Caco-2 cells. After the cells were adherent, 100 μL/well of various concentrations of Gen-NH_2_-MCM-41@SA (50 μM, 25 μM, 12.5 μM, 6.25 μM, 3.125 μM, 1.57 μM, 0.78 μM, and 0.39 μM) was added to the cells. DMEM medium with the same volume as Gen-NH_2_-MCM-41@SA was added to the normal group. Six parallel wells were set for each concentration in the experimental group, and six wells were set for the solvent control group. Caco-2 cells with logarithmic and long-term growth were taken and made into a cell suspension after trypsin digestion and wall removal. After counting, they were diluted into the cell suspension with high-sugar DMEM medium containing 10% FBS, and 100μL of the cell suspension was added to each well of the 96-well culture plate, with each well containing 6000-cell suspensions. After the cells adhered to the wall, 100 μL of the above-prepared sample solution was added to the wells, and the same volume of DMEM medium was added to the control wells [13].

After 72 h, the solution was removed, and cells were washed with PBS twice. Then, 10 μL of CCK-8 (5 mg/mL) solution was added to each well, and the cells were cultured in a 37 °C, 5% CO_2_ incubator for 4 h. The detection wavelength was set at 450 nm, and the optical density (OD) value of each hole was measured [34].
$$\mathrm{cell}\;\mathrm{viability}=\frac{A_{\mathrm{experimental}\;\mathrm{group}}-A_{\mathrm{RV}\;\mathrm{blank}\;\mathrm{group}\;}}{A_{\mathrm{control}\;\mathrm{group}}-A_{\mathrm{RV}\;\mathrm{blank}\;\mathrm{group}\;}}\times 100\%$$

#### 2.5.4. Anti-RV Effects of Gen-NH_2_-MCM-41@SA Nanoparticles

The solubility of Gen in water is 8.7 μg/mL, so the maximum concentration of Gen in this module was selected as 40 μm, and the anti-RV effect was verified by observing CPE combined with the therapeutic index (TI) [11].

##### Inhibition of RV Attachment of Gen-NH_2_-MCM-41@SA Nanoparticles and Gen

Gen-NH_2_-MCM-41@SA and Gen (5 μM, 10 μM, 20 μM, 30 μM, and 40 μM) were added to the 96-well culture plate of Caco-2 cells grown to a monolayer, which was repeated for 6 wells for each solution concentration, adding 100 μL to each well. The positive control group was mixed with an equal volume of ribavirin, and the negative control group was mixed with an equal volume of DMEM medium without FBS and then incubated in a 37 °C incubator with 5% CO_2_ for 2 h. Then, the drug solution was removed and discarded. To the remaining wells, 100 μL of 100 TCID50 WA-RV virus (the virus was incubated with 10 μg/mL pancreatin at 37 °C for 30 min) was added and incubated in an incubator at 37 °C with 5% CO_2_ for 2 h. Then, the virus was removed, and 200 μL of cell maintenance solution was added to each well. The cultures were placed at 37 °C with 5% CO_2_ in an incubator for continuous incubation and observation. After continuous culture for 48 h, 10 μL of CCK-8 solution (5 mg/mL) was added to each well, and the cells were placed in an incubator with 5% CO_2_ at 37 °C for 4 h. The detection wavelength was set at 450 nm on an enzyme-conjugate immunodetector, and the light absorption values of each well were determined [35].
$$\mathrm{Virus}\;\mathrm{inhibition}\;\mathrm{rate}=\frac{A_{\mathrm{experimental}\;\mathrm{group}}-A_{\mathrm{RV}\;\mathrm{blank}\;\mathrm{group}\;}}{A_{\mathrm{control}\;\mathrm{group}}-A_{\mathrm{RV}\;\mathrm{blank}\;\mathrm{group}\;}}\times 100\%$$
$$TI=\frac{TC50}{EC50}$$
where *TI* is the treatment index; *TC*50 is 50% cytotoxic concentration; and *EC*50 is half effective concentration.

##### Effects of Gen-NH_2_-MCM-41@SA Nanoparticles and Gen on the Direct Inactivation of RV

The 96-well culture plate of Caco-2 cells grown to monolayer was taken out, the old culture medium was discarded, and different concentrations of Gen-NH_2_-MCM-41@SA and Gen solutions (5 μM, 10 μM, 20 μM, 30 μM, and 40 μM) were added to the incubator at 37 °C with 5% CO_2_ for 2 h. DMEM culture medium without FBS was washed once, and 200 μL of 100TCID50 disease toxin was added to each well (RV-WA and 10 μg/mL trypsin were treated at 37 °C for 30 min) [36]. In the control group, only an equal volume of DMEM was added, and the disease toxin was removed after 2 h of culture. After adding DMEM without FBS, the culture was continued for 48 h. Then, 10 μL of CCK-8 solution (5 mg/mL) was added to each well, and the culture was continued for 4 h in an incubator at 37 °C and 5% CO_2_. The wavelength of the microplate reader was set at 450 nm, and the light absorption values of each well were determined [37].

##### Effects of Gen-NH_2_-MCM-41@SA Nanoparticles and Gen on Inhibiting RV Replication

The 96-well culture plate of Caco-2 cells grown to a monolayer was removed, the old medium was carefully removed, and the culture medium was washed with DMEM medium without FBS. Then, 200 μL of 100 TCID50 RV-WA toxin was added to each well (the virus was treated with 10 μg/mL trypsin at 37 °C for 30 min). In the normal control group, only the same volume of DMEM was added, and then the cells were placed in a 37 °C and 5% CO_2_ incubator for 2 h. Then, the disease toxin was removed, and DMEM culture medium without FBS was gently rinsed. Gen-NH_2_-MCM-41@SA and Gen solutions at different concentrations (5 μM, 10 μM, 20 μM, 30 μM, and 40 μM) were added, and cells were incubated in an incubator at 37 °C with 5% CO_2_ for 24 h. Next, 10 μL of CCK-8 solution (5 mg/mL) was added to each well, and cells were cultured in an incubator with 5% CO_2_ at 37 °C for 4 h. The detection wavelength was set at 450 nm for enzyme-conjugate immunodetection, and the light absorption values of each well were determined [38].

### 2.6. Administration of Gen-NH_2_-MCM-41@SA and Gen against RV Infection in Neonatal Mice

#### 2.6.1. Culture of Suckling Mice of Kunming Species

Pregnant Specific Pathogen-Free Kunming mice were fed at approximately 20–25 °C with a relative humidity of 40–70%. Kunming mice in the last trimester of pregnancy were separated before giving birth naturally. Newborn neonatal mice were weighed after birth, and those with significant individual differences were excluded. The number of neonatal mice per cage was controlled at 9–10 [39].

#### 2.6.2. Study on Modeling and Drug Administration in Kunming Species Mice

After the Kunming mice were 4 days old, 200 µL of DMEM or RV-SA at a virus titer of 10^−4.64^ TCID50/mL was administered to them via oral gavage. After gavage, the neonatal mice were placed back with their mothers and fed normal breastmilk for three consecutive days [40,41].

According to the random distribution scheme, Kunming mice (4 days old) were divided into nine groups: (1) normal group, (2) RV group, (3) low-dose Gen group (0.075 mg/g), (4) medium-dose Gen group (0.15 mg/g), (5) high-dose Gen group (0.3 mg/g), (6) low-dose Gen-NH_2_-MCM-41@SAgroup (0.075 mg/g), (7) medium-dose NH_2_-MCM-41@SA group (0.15 mg/g), (8) high-dose Gen-NH_2_-MCM-41@SA group (0.3 mg/g), and (9) ribavirin group (0.1 mg/g) (Furen Pharmaceutical Group, Batch No. 1210011). Mice in the normal group were given DMEM medium by gavage. Mice in the RV and ribavirin groups and each Gen and Gen-NH_2_-MCM-41@SA dose group were given 200 µL of RV by gavage. Then, 200 µL of Gen or Gen-NH_2_-MCM-41@SA was given to the mice in each Gen and Gen-NH_2_-MCM-41@SA dose group for 2 days post-infection. DMEM medium was given to the mice in the RV group and normal group, and 0.1 mg/g ribavirin was given to the ribavirin control group once a day for 5 days. The clinical signs of each group were observed and recorded daily after infection: mental status, dehydration, redness of the anus, and the number of diarrhea cases in the newborn mice. The diarrhea rate was calculated by the following formula: diarrhea rate = the number of neonatal mice with diarrhea/the total number of neonatal mice [42]. The degree of diarrhea was judged and scored on a scale of 0–4 based on the color and shape of the feces. Meanwhile, we collected feces of newborn mice to detect RV antigen by the colloidal gold method. The operation was carried out strictly according to the kit instructions [43].

#### 2.6.3. HE Staining of Histopathological Sections

Three mice in each group were sacrificed on day 3 after administration, and two small tissue pieces were taken from the small intestine of each suckling mouse. The tissue pieces were rinsed with PBS, fixed with 4% paraformaldehyde, and subjected to routine HE staining.

### 2.7. Statistical Analysis

SPSS 13.0 software was used for data analysis. The one-way ANOVA method was used to compare the mean OD values of the drug group and the model group. The PROBIT regression method was used to perform regression analysis on drug concentration, cell survival rate, drug concentration, and virus inhibition. TC_50_ and IC_50_ were obtained, and the treatment index TI was calculated. TI > 2 is considered effective, and the larger the TI, the greater the safety range of the drug.

## 3. Results and Discussion

### 3.1. Physico-Chemical Characterization

The general properties of the R/silica complexes were characterized by XRD, DSC, SEM, Zeta potential, FT-IR, and BET analyses.

#### 3.1.1. X-ray (Small Angle) Diffraction

Low-angle XRD patterns of the prepared MCM-41 and NH_2_-MCM-41 samples are shown in Figure 5.

We can see the characteristic diffraction peaks of the two crystal planes, which are (110) and (200).

The characteristic diffraction peak of MCM-41 fully conforms to the characteristic diffraction peak of MCM-41, indicating that the synthesized sample had a long-range ordered hexagonal phase channel structure. It can be found from the characteristic diffraction peak of the crystal plane that with the addition of APTES and with the increase in the added amount, the number of introduced amino groups increased, and the influence on the crystal structure in the process of copolycondensation was also greater.

The intensity of the diffraction peak is smaller, and the characteristic peaks of crystal planes (110) and (200) gradually weaken and disappear, but remnants of some peaks can still be seen. The intensity of the diffraction peak represents the degree of order of the material, indicating that with the addition of APTES, the order degree of NH_2_-MCM-41 was affected. With the increase in the APTES dosage, the order degree decreased, but it still maintained the original hexagonal channel structure.

#### 3.1.2. Nitrogen Adsorption–Desorption (BET) Analysis of MCM-41 and NH_2_-MCM-41

Compared with the six standard adsorption isotherms classified by IUPAC, the nitrogen adsorption–desorption isotherm results in Figure 6A correspond to the Langmuir IV hysteretic adsorption isotherm and H4. According to the capillary condensation phenomenon formed by the hysteresis loop, the characteristics of the material pore structure can be judged. The above figure shows that the channels of NH_2_-MCM-41 are crack holes. It can be seen in Table 1 that the pore size distribution and specific surface area of MCM-41 are much larger than those of NH_2_-MCM-41. The main reason is that the addition of APTES increased the diameter of micelles in the copolycondensation process, which is also one of the characteristics of the copolycondensation method.

Although the reduced specific surface area and reduced pore size have limitations for the drug delivery process, they do not cause drug leakage and are more effective for closing the pores, and the nanoparticles are more acid–base-dependent when sealed with SA.

#### 3.1.3. Fourier Infrared Spectroscopy (FT-IR) Analysis of MCM-41 and NH_2_-MCM-41

By comparing the infrared spectra of a and B, the main functional groups and changes in the compound and the values of some vibrational absorption peaks in the synthesis process were determined.

Figure 7 shows the infrared spectra of MCM-41 and NH_2_-MCM-41. The black line represents MCM-41, and the red line represents NH_2_-MCM-41. The peak at 786 cm^−1^ indicates the existence of symmetric stretching vibration of Si-O-Si, the peak at 1060 cm^−1^ indicates the existence of antisymmetric stretching vibration of Si-O-Si, and the peak at 980 cm^−1^ indicates the existence of bending vibration of Si-O; peaks at 2925 cm^−1^ and 2852 cm^−1^ are the stretching vibration of methylene. The peak at 3425 cm^−1^ shows that there are stretching vibrations of hydroxyl and amino groups, and the bending vibration of NH_2_ is indicated by the peak at 1470 cm^−1^, indicating that the amino functional groups were successfully copolycondensated onto MCM-41; that is, the synthesis of aminated NH_2_-MCM-41 was successful.

#### 3.1.4. Particle Size Analysis and Zeta Potential Measurement

In Figure 8, a is the particle size of Gen-NH_2_-MCM-41 nanoparticles, about 320.2 nm, and b is Gen-NH_2_-MCM-41 encapsulated by Gen-NH_2_-MCM-41@SA. The particle size of nanoparticles is about 380.3 nm. It can be seen that the particle size increased significantly after SA encapsulated the particles, and the b particles are evenly dispersed in the figure, indicating that the SA encapsulation effect met expectations. It can be seen in Figure 9A that the color of the whole nanoparticles changed significantly after being encapsulated by SA. Gen-NH_2_-MCM-41 can be seen in the electron microscopy results in Figure 9B. The structure of Gen-NH_2_-MCM-41@SA nanoparticles is nearly round, and the appearance is excellent. In addition, the shape is nearly round or spherical with a stable character. It is suitable as the shell or substrate of the drug-loading model.

#### 3.1.5. SEM Analysis of MCM-41 and NH_2_-MCM-41

Figure 10 shows the SEM images of MCM-41 and NH_2_-MCM-41 ((a) MCM-41 and (b) NH_2_-MCM-41). It can be seen in the two images that mesoporous silica is hexagonal prismatic with a uniform particle distribution and regular shape. The addition of APTES had an effect on the morphology of MCM-41 during the copolycondensation process. However, the effect was small, so it is suitable as a drug delivery substrate.

#### 3.1.6. Differential Scanning Thermal Analysis (DSC)

It can be seen in the DSC results that the melting absorption peak of simple Gen API is 89.1 °C, indicating that Gen API has crystal characteristics (As Shown in the Figure 11). The absorption peak of Gen-NH_2_-MCM-41 shifted forward, and the physical mixture of Gen-NH_2_-MCM-41 + SA still had two relatively split absorption peaks. In contrast, for Gen-NH_2_-MCM-41 heated at the same heating rate as Gen-NH_2_-MCM-41@SA, the disappearance of the characteristic absorption peak of the nanoparticles indicates that the crystal form of the nanoparticles was further changed, which is different from the results of the simple physical mixing of Gen-NH_2_-MCM-41 + SA, which proves that SA encapsulation was successful.

#### 3.1.7. Adsorption Performance and Drug Loading

The adsorption capacity and drug loading of Gen and NH_2_-MCM-41 in different proportions calculated according to Formulas (1) and (2) are shown in Table 2. It can be seen in the table that with the increase in NH_2_-MCM-41, the adsorption capacity of Gen gradually tended to become saturated. The adsorption of Gen by NH_2_-MCM-41 was mainly physical adsorption caused by its porous structure, and methylene was also introduced into mesoporous silica after amination. The hydrophobic effect of methylene can also improve the drug loading of Gen.

#### 3.1.8. Gen-NH_2_-MCM- 41@SA Analysis of Sustained-Release Properties of Nanoparticles In Vitro

From the UV absorption of Gen (as shown in Figure 12 below), it can be seen that Gen-NH_2_-MCM-41 without SA reached the top of the drug release curve within 2 h, and the cumulative release reached more than 90% within 3 h, while Gen-NH_2_-MCM-41@SA nanoparticles had little or only a small amount of drug release in simulated gastric juice. When the rotating basket was replaced with simulated intestinal fluid, Gen-NH_2_-MCM-41@SA nanoparticles were released rapidly and reached the peak of the release curve at 6 h. The comparison showed that the nanoparticles coated with SA had a sustained-release performance.

#### 3.1.9. Gen-NH_2_-MCM-41@SA Drug Release Model Fitting under Different pH Conditions

The zero-order model was used to explain the dissolution-controlled drug release (as shown in Table 3 below). The first-order model and Higuchi model are the classical models of drug plane diffusion. At present, the Korsmeyer–Peppas model is the most widely used in drug release, which combines dissolution and diffusion. When the release index *n* < 0.45, drug release follows the Fickian diffusion mechanism, presenting Fick diffusion; when 0.45 < *n* < 0.89, drug release follows the non-Fickian diffusion mechanism, which is manifested as the synergistic effect of dissolution and diffusion. When *n* > 0.89, the drug release follows Super Case II transport, which shows dissolution. As can be seen from the drug release fitting results under different pH conditions in the table, Gen-NH_2_-MCM-41@SA nanoparticles fit Korsmeyer–Peppas better, and when pH = 1.0, *n* = 0.25124; when pH = 6.8, *n* = 0.27874; when pH = 7.4, *n* = 0.41211, indicating that the release index n is less than 0.45 in different pH environments, indicating that the release of Gen-NH_2_-MCM-41@SA nanoparticles occurs via Fick diffusion.

In conclusion, as can be seen from the UV absorption of Gen, Gen-NH_2_-MCM-41 without SA reached the peak of the drug release curve within 2 h, and the cumulative release reached more than 90% within 3 h, while Gen-NH_2_-MCM-41@SA nanoparticles had little or no drug release in simulated gastric juice. When the simulated intestinal fluid replaced the gastric juice in the spinning bucket, Gen-NH_2_-MCM-41@SA nanoparticles were released rapidly and reached the peak of the release curve at 6 h. According to the fitting of release conditions, Gen-NH_2_-MCM-41@SA nanoparticles conform to the Korsmeyer–Peppas model and Fick diffusion. This is in line with our expectations.

### 3.2. In Vitro Cell Experiment

#### 3.2.1. CCK-8 Detected the Cytotoxicity of NH_2_-MCM-41, SA, and Gen in Caco-2

Caco-2 cells were cultured for 12 h and treated with NH_2_-MCM-41, SA, and Gen at concentrations of 0.78 μM, 1.56 μM, 3.12 μM, 6.25 μM, 12.5 μM, 25 μM, 50 μM, and 100 μM. After 48 h, cell viability was detected by the CCK-8 method. As the results show in Figure 13, SA showed no significant toxicity in Caco-2 cells at concentrations below 100 μM, and the cell survival rate was above 95%. When the concentration was 100 μM, Gen showed slight toxicity in Caco-2 cells, and the cell survival rate was 80.34%. Therefore, when exploring the activity of Gen-NH_2_-MCM-41@SA nanoparticles, the relative concentration of Gen should be controlled below 100 μM. NH_2_-MCM-41 showed no obvious toxicity in Caco-2 cells at a concentration of 100 μM, which proved that the three materials were safe for cells.

#### 3.2.2. Gen-NH_2_-MCM-41@SA and Gen Anti-RV-WA Adsorption

The adsorption results of Gen-NH_2_-MCM-41@SA and Gen against RV-WA are shown in Figure 14. There was no significant difference in the inhibition rate in the Gen group. The maximum inhibition rate was 5.58%, and the TI value was 1.05 when the concentration was 10 μM. There was no significant difference in the inhibition rate within the Gen-NH_2_-MCM-41@SA group (*p* > 0.05). When the concentration was 30 μM, the maximum inhibition rate of Gen-NH_2_-MCM-41@SA was 20.26%, and the TI value was 1.81. It is generally believed that TI ≥ 2 proves that the drug has antiviral activity. It can be concluded that Gen and Gen-NH_2_-MCM-41@SA nanoparticles have no anti-RV-WA adsorption effect.

#### 3.2.3. Gen-NH_2_-MCM-41@SA and Gen Can Directly Inactivate RV-WA

The results of the inactivation of RV-WA by Gen-NH_2_-MCM-41@SA and Gen are shown in Figure 15. The inhibition rate was 19.26%, and the TI value was 2.34 in the Gen group at 40 μM, and the difference was statistically significant *(p* < 0.05). The inhibition rate was 34.59%, and the TI value was 3.56 in the Gen-NH_2_-MCM-41@SA group at 40 μM. There was a significant difference between the 20 μM, 30 μM, and 40 μM drug concentration groups. The TI values of the two groups showed that both groups had the effect of killing RV-WA, but the activity of the Gen-NH_2_-MCM-41@SA group was more substantial.

#### 3.2.4. Gen-NH_2_-MCM-41@SA and Gen against RV-WA Biosynthesis

The anti-RV-WA biosynthesis results of Gen-NH_2_-MCM-41@SA and Gen are shown in Figure 16. The inhibition rate and the TI value of the Gen group at 40 μM were 34.63% and 2.89, respectively, with statistical significance (*p* < 0.05). At a concentration of 30 μM, the inhibition rate of the Gen-NH_2_-MCM-41@SA group was 62.24%, and the TI value was 5.8. There were significant differences between the 5 μM, 10 μM, 20 μM, 30 μM, and 40 μM drug concentration groups. The TI values of the two groups showed that both groups had anti-RV-WA biosynthesis effects, but the activity of the Gen-NH_2_-MCM-41@SA group was nearly twice as high as that of the Gen group.

### 3.3. Effect of Gen-NH_2_-MCM-41@SA Nanoparticles on RV-SA Infection in Suckling Mice

#### 3.3.1. Changes in Body Weight of Suckling Mice after Drug Administration

P.S: The experimental groups were the H-gen group (0.3 mg/g), M-Gen group (0.15 mg/g), L-Gen group (0.075 mg/g), H-Gen-NH_2_-MCM-41@SA group (0.3 mg/g), M-Gen-NH_2_-MCM-41@SA group (0.15 mg/g), and L-Gen-NH_2_-MCM-41@SA group (0.075 mg/g).

As can be seen in Figure 17, the weight gain rate of the H-Gen-NH_2_-MCM-41@SA group was closer to that of the positive control drug (ribavirin) group, and the weight gain rates of the M-Gen-NH_2_-MCM-41@SA and L-Gen-NH_2_-MCM-41@SA groups were also higher than that of the model group. This shows that RV-SA had a specific therapeutic effect on diarrhea in suckling mice, but the effect was slightly lower than that of the H-Gen-NH_2_-MCM-41@SA group.

#### 3.3.2. Diarrhea Score of Suckling Mice after Administration and Diarrhea Degree after RV-SA Infection in Suckling Mice

The scores for diarrhea and feces of suckling mice are shown in Figure 18, and the score grade ranges from 0~4: 0 is no feces; 1 is normal brown stools; 2 is light yellow soft stool; 3 is divided into watery yellow loose stools; and 4 is watery stool. The weight and diarrhea rate of suckling mice were recorded daily.

From the perspective of diarrhea recovery, the diarrhea recovery of nursing mice after administration was better than that of the model group (as shown in Figure 19). The comparison showed that the score of the H-Gen-NH_2_-MCM-41@SA group decreased the most, almost similar to that of the positive control drug (ribavirin) group, indicating that the treatment effect on the H-Gen-NH_2_-MCM-41@SA group was good.

#### 3.3.3. Pathological Changes in Intestinal Tissue on the Third Day after Administration

On the third day after the treatment, the results were compared, and small parts of normal epithelial cells were seen to show an apparent polarization phenomenon (as shown in Figure 20), epithelial cell swelling, loose light dye in the cytoplasm (black arrow), and a small circular cavity (red arrows). With RV infection-SA, epithelial cells showed noticeable pathological changes, a portion of the intestinal villus was swollen, and fluffy contours gradually became clear. Intestinal wall edema was prominent, cell integrity was destroyed, and the lesions were significantly improved after administration. Compared with the H-Gen group, the intestinal epithelial cells in the H-Gen-NH_2_-MCM-41@SA group recovered better, but certain vacuolar changes were still observed in the L-Gen and L-Gen-NH_2_-MCM-41@SA groups.

## 4. Conclusions

Using aminated mesoporous silica technology, Gen was successfully encapsulated to prepare the nucleus of the nanoparticles. SA was used as the shell to encapsulate the fundamental particles to form a core–shell composite structure, and Gen-NH_2_-MCM-41@SA nanoparticles were synthesized. Gen-NH_2_-MCM-41@SA nanoparticles were successfully prepared by the copolycondensation method. Meanwhile, the application of SA is essential for the selection of encapsulation agents.

In vitro stimulation experiments showed that the drug release process was pH-responsive. The experimental results of WA-RV infection in Caco-2 cells showed that Gen-NH_2_-MCM-41@SA nanoparticles could directly kill RV and exert anti-RV biosynthesis effects. Gen-NH_2_-MCM-41@SA nanoparticles had a therapeutic effect on diarrhea induced by SA-RV infection in neonatal mice, which could reduce the diarrhea rate and improve the diarrhea degree in neonatal mice, and the effect of the high-dose group (0.3 mg/g) was more pronounced.

Compared with Gen, the water-insoluble condition of Gen-NH_2_-MCM-41@SA nanoparticles was effectively improved, providing a new preparation idea for similar water-insoluble monomers of natural drug monomers. The results of infection model establishment in suckling mice showed that the L-Gen group, M-Gen-NH_2_-MCM-41@SA group, and H-Gen-NH_2_-MCM-41@SA group had better efficacy in terms of the diarrhea rate and diarrhea index, and intestinal tissue recovered well in histopathological sections. Further, Gen-NH_2_-MCM-41@SA nanoparticles could effectively inhibit RV-WA virus replication and had better efficacy.

Aiming at the deficiency that Gen is difficult to dissolve in water, this study used NH_2_-MCM-41 nanoparticles as the carrier to encapsulate whole nanoparticles by using the principle of valence bond adsorption between the -NH_2_ group and SA surface -COOH. In addition, SA can further enhance the pH responsiveness of whole nanoparticles and target Gen to the small intestine. To achieve the purpose of killing rotavirus, a core–shell composite structure, with Gen-NH_2_-MCM-41 as the core and SA as the shell, was finally formed. This structure not only has good stability but also can effectively prevent the agglomeration of nanoparticles so as to achieve complementary advantages between the core and shell.

## Figures and Tables

**Figure 1 pharmaceutics-14-01337-f001:**
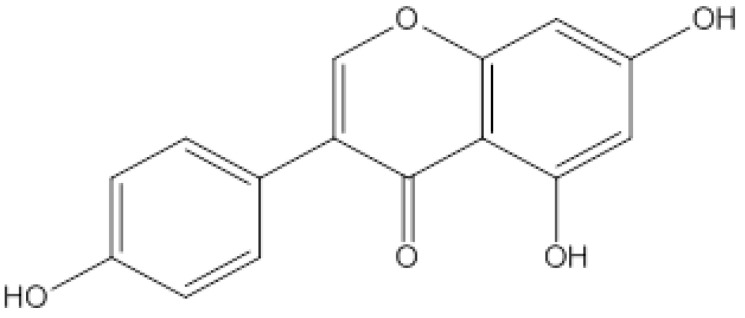
The chemical structural formula for Gen.

**Figure 2 pharmaceutics-14-01337-f002:**
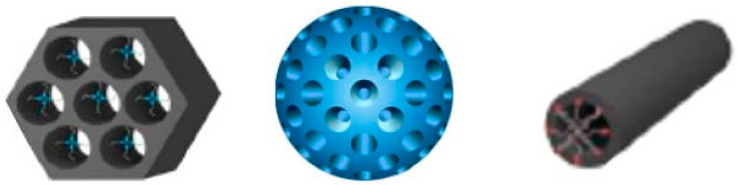
Various morphological models of mesoporous silica.

**Figure 3 pharmaceutics-14-01337-f003:**
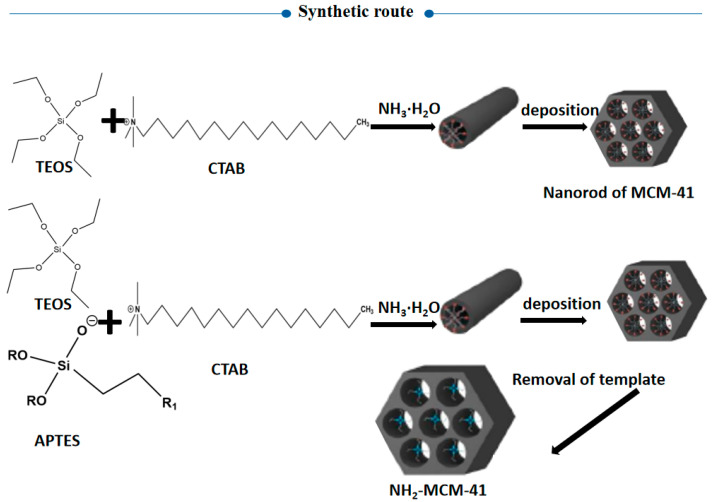
Synthesis route of aminated mesoporous silicon.

**Figure 4 pharmaceutics-14-01337-f004:**
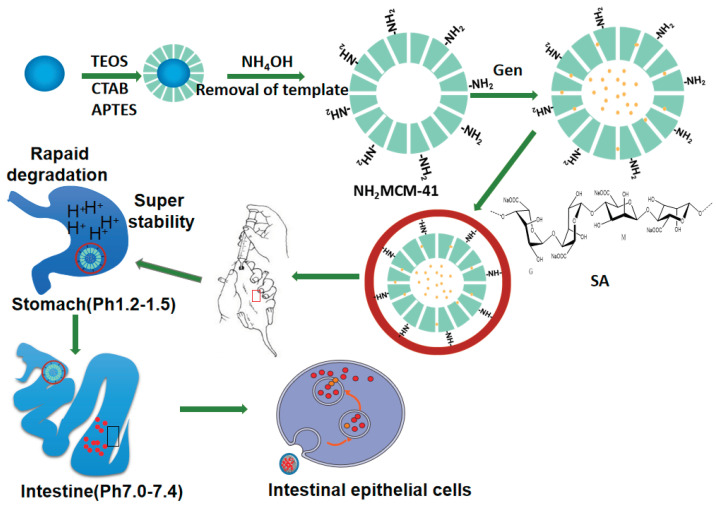
The synthesis process of NH_2_-MCM-41@SA nanoparticles.

**Figure 5 pharmaceutics-14-01337-f005:**
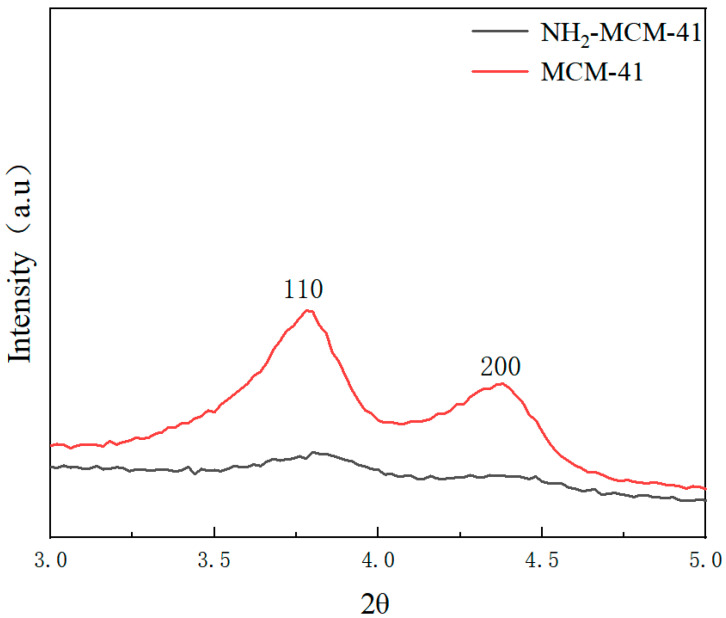
XRD patterns of MCM-41 (red) and NH_2_-MCM-41 (black).

**Figure 6 pharmaceutics-14-01337-f006:**
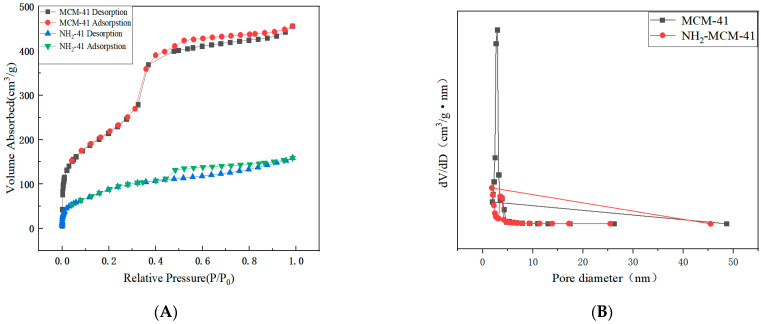
N_2_ adsorption–desorption isotherms (**A**) and pore size distribution (**B**) of MCM-41 and NH_2_-MCM-41.

**Figure 7 pharmaceutics-14-01337-f007:**
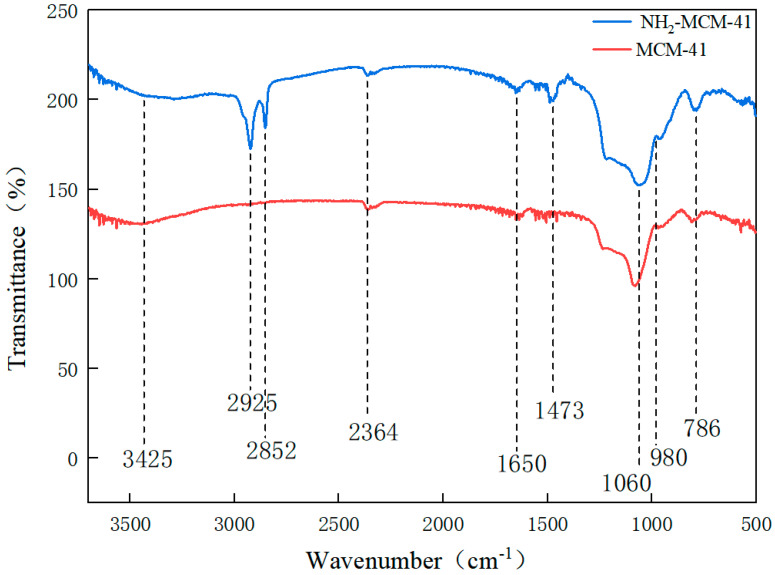
FT-IR diagram of MCM-41 and NH_2_-MCM-41.

**Figure 8 pharmaceutics-14-01337-f008:**
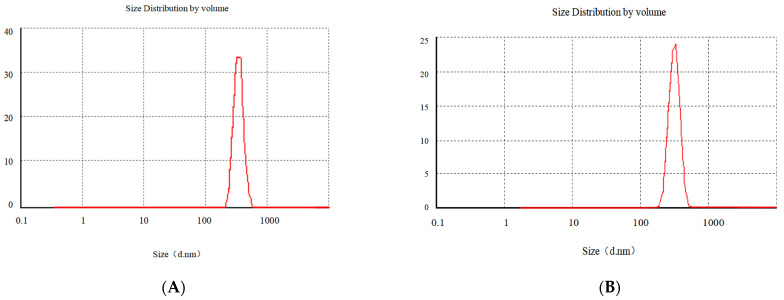
The particle size of Gen-NH_2_-MCM-41 (**A**) and Gen-NH_2_-MCM-41@SA nanoparticles (**B**).

**Figure 9 pharmaceutics-14-01337-f009:**
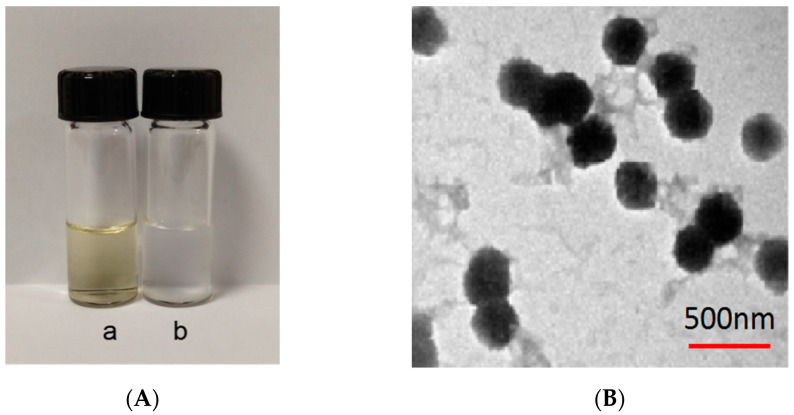
(**A**) a: Gen-NH_2_-MCM-41; b: Gen-NH_2_-MCM-41@SA nanoparticles; (**B**) electron microscopy of Gen-NH_2_-MCM-41@SA nanoparticles.

**Figure 10 pharmaceutics-14-01337-f010:**
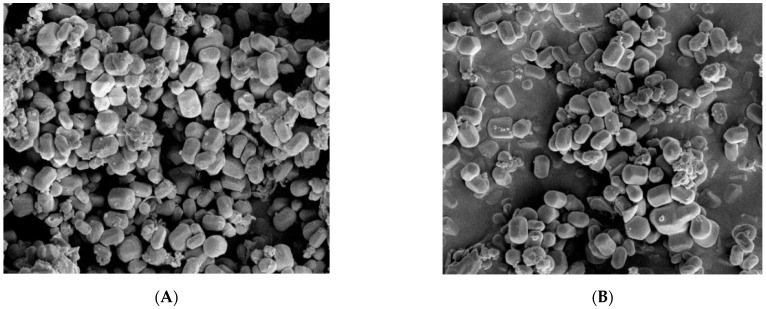
SEM images of MCM-41 (**A**) and NH_2_-MCM-41 (**B**).

**Figure 11 pharmaceutics-14-01337-f011:**
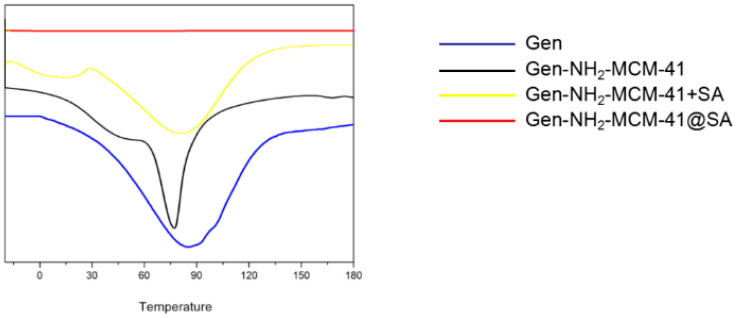
DSC results of Gen, Gen-NH_2_-MCM-41, Gen-NH_2_-MCM-41 + SA physical mixture, and Gen-NH_2_-MCM-41@SA nanoparticles.

**Figure 12 pharmaceutics-14-01337-f012:**
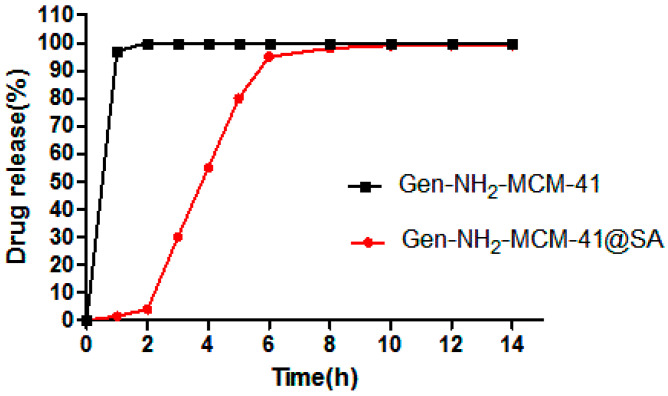
Gen-NH_2_-MCM-41 and Gen-NH_2_-MCM-41@SA in vitro drug release curve of nanoparticles.

**Figure 13 pharmaceutics-14-01337-f013:**
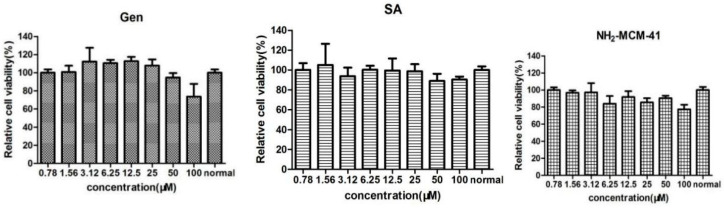
The effects of NH_2_-MCM-41, SA and Gen on the activity of Caco-2 cells.

**Figure 14 pharmaceutics-14-01337-f014:**
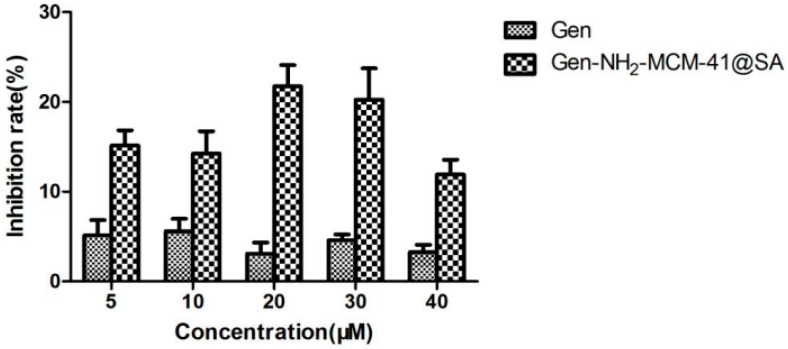
Gen-NH_2_-MCM-41@SA and Gen inhibited the attachment and replication of (RV) strain Wa in Caco-2 cells.

**Figure 15 pharmaceutics-14-01337-f015:**
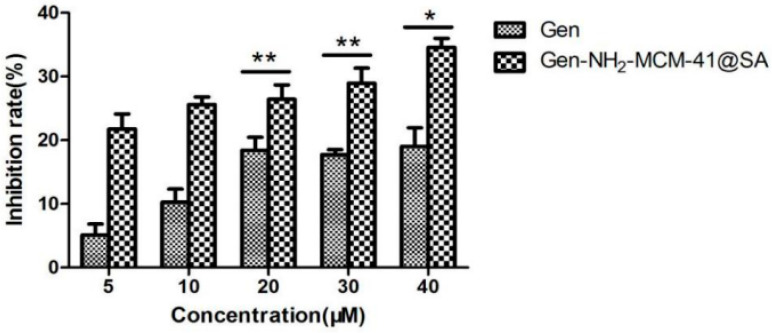
Gen-NH_2_-MCM-41@SA and Gen inactivation RV-Wa (* *p* < 0.05, ** *p* < 0.01).

**Figure 16 pharmaceutics-14-01337-f016:**
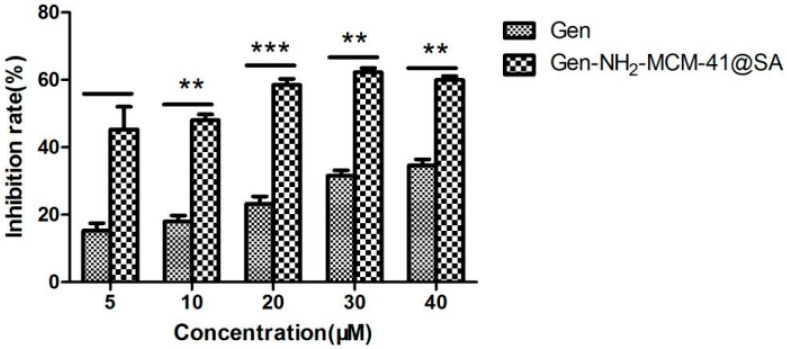
Anti-RV biosynthesis activity of Gen-NH_2_-MCM-41@SA and Gen (** *p* < 0.01, *** *p*< 0.001).

**Figure 17 pharmaceutics-14-01337-f017:**
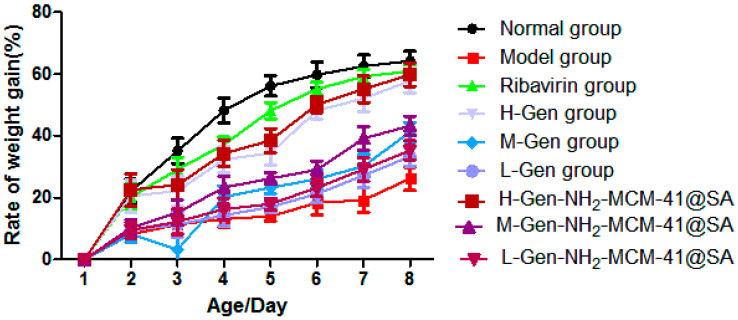
The body weight change in suckling mice in different groups.

**Figure 18 pharmaceutics-14-01337-f018:**
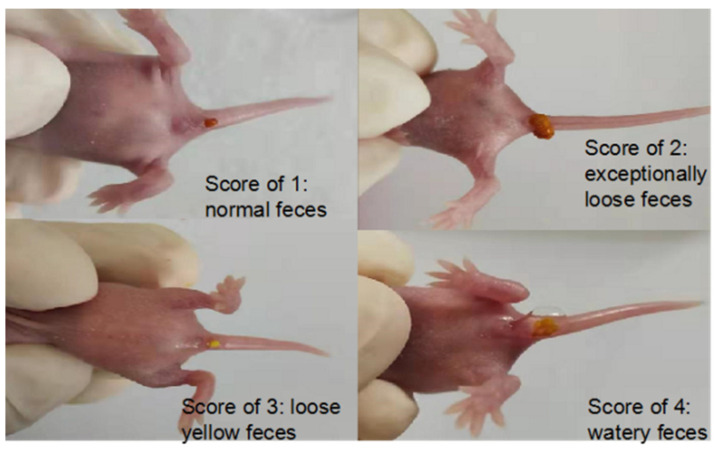
Diarrhea in suckling mice.

**Figure 19 pharmaceutics-14-01337-f019:**
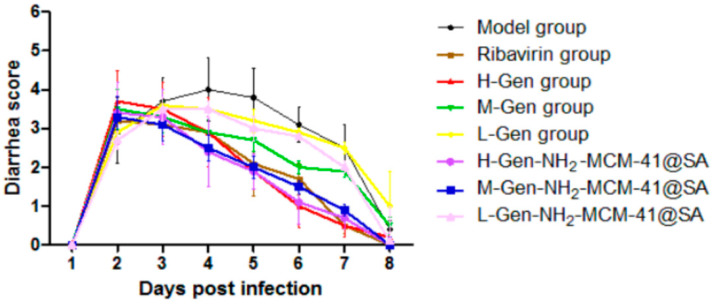
Diarrhea score of suckling mice in different groups.

**Figure 20 pharmaceutics-14-01337-f020:**
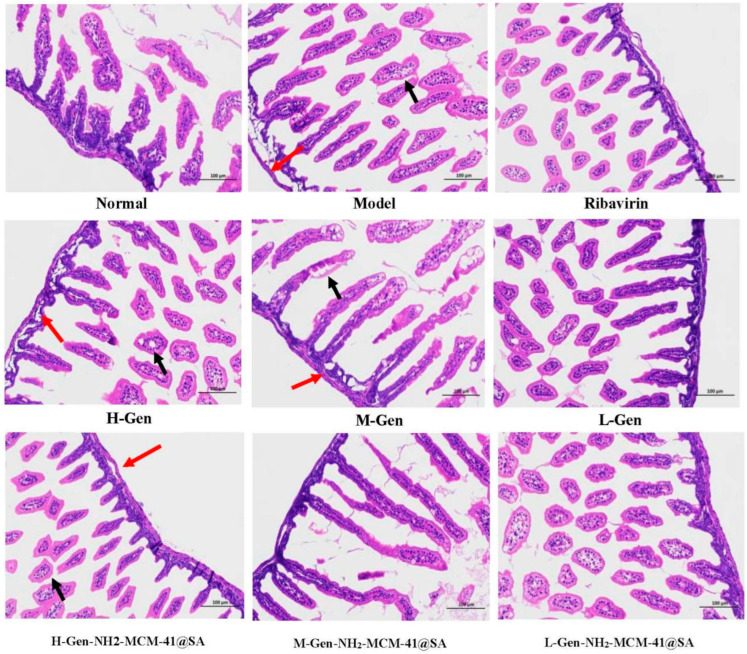
Cross-section of small intestine tissue of different groups of suckling mice on the third day after administration. (red arrows: a few rounded vacuoles can be seen; black arrows: cytoplasm loose).

**Table 1 pharmaceutics-14-01337-t001:** The pore structural parameters of MCM-41 and NH_2_-MCM-41.

Sample Name	Specific Surface Area (m^2^/g)	Pore Volume (cm^3^/g)	Aperture (nm)
MCM-41	804	0.795	2.95
NH_2_-MCM-41	452	0.408	1.88

**Table 2 pharmaceutics-14-01337-t002:** The adsorption capacity and drug loading of Gen and NH_2_-MCM-41 in different proportions.

Gen: NH_2_-MCM-41	Adsorption Capacity (mg/g ± SD)	Drug Loading (mg/g ± SD)
1:0.5	13.15 ± 0.89%	12.65 ± 1.53%
1:1	9.14 ± 0.71%	8.97 ± 1.28%
1:2	8.1 ± 1.76%	6.96 ± 1.10%

**Table 3 pharmaceutics-14-01337-t003:** Fitting of Gen-NH_2_-MCM-41@SA nanoparticle drug release model under different pH conditions.

Pharmacokinetic Model	Fitted Equation		pH	Correlation Coefficient/*R*^2^
Zero-order	$\frac{M_{t}}{M_{\infty }}$	0.00272t + 0.4758	1.0	0.61442
		0.00285t + 0.05282	6.8	0.69925
		0.001475t + 0.00122	7.4	0.71021
First-order	$\ln(1-\frac{M_{t}}{M_{\infty }})$	−0.0049t − 0.1251	1.0	0.64511
		−0.00412t − 0.11241	6.8	0.7945
		−0.000421t − 0.00714	7.4	0.8147
Higuchi	$\frac{M_{t}}{M_{\infty }}$	0.00659t^1/2^ + 0.14915	1.0	0.62584
		0.00124t^1/2^ + 0.01548	6.8	0.61591
		0.0000987t^1/2^ + 0.00126	7.4	0.79948
Korsmeyer–Peppas	$\frac{M_{t}}{M_{\infty }}$	0.1301t^0.25124^	1.0	0.98101
		0.09801t^0.27874^	6.8	0.92156
		0.00051402t^0.41211^	7.4	0.95641

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
