# Peer review of "The Preparation of Gen-NH2-MCM-41@SA Nanoparticles and Their Anti-Rotavirus Effects"

_pharmaceutics, 2022, doi:10.3390/pharmaceutics14071337_

Round 1

Reviewer 1 Report

Manuscript ID: pharmaceutics-1729020 titled “The preparation of pH-sensitive Gen-NH2-MCM-41@SA nanoparticles and its effect on anti-rotavirus” submitted to topic -Microfluidics Applied in Nanomedicine and Pharmaceutics, by Wenchang Zhao et al discusses the potential of Gen-NH2-MCM-41@SA nanoparticles, as drug carrier vehicles for improvement of solubility and poor absorption, and the possible application of isoflavone genistein in the clinical treatment of RV infection based on pH sensitive release of genistein.

General comment-

Change in title is recommended- The preparation of pH-sensitive Gen-NH2-MCM-41@SA nano-2 particles and its anti-rotavirus effect

Introduction- The Gen abbreviation is not routinely used in literature. Therefore at the first instance the proper common name should be mentioned. Further on the abbreviated form can be used. Please use capital letter G in the sentences to indicate abbreviated form (Gen). Correct this throughout the manuscript

Introduction does not indicate references 2-3 and directly starts at ref 1& 4. The bibliography 1-4 are unrelated references for the context being discussed. Again ref 5 is missing in text, but mention of 6,7, which also seem unrelated to the context discussed. Further the references 9 and 10, 11, 12 & 13 are also not related to the context being discussed.  The entire paper has only 13 references, which is highly unlikely. This needs revision.

Figure 7- the legend for yellow graph should include “Gen-NH2-MCM-41+SA physical mixture”

In 3.2. In vitro cell experiment section, it was mentioned that CCK-8 detected the cytotoxicity of NH2-MCM-41, SA and Gen on Caco-2. The concentration of SA and Gen at the mentioned values can be calculated by their molecular weight. How/what was the MW of NH2-MCM-41 considered for dilutions?

Please insert DOI for all the references in the references section.

Author Response

1. The title has been revised

2.The references have been enriched

3.DSC results have been modified legend

4. Calculate the molecular weights of the three according to the mixing ratio of the three materials

5 DOI values have been inserted

Reviewer 2 Report

Dear respected editor

The manuscript entitled " Formulation of Isopropyl isothiocyanate loaded nanoliposomes delivery systems: in-vitro characterization and in-vivo 3 assessment" described the preparation of genistein into pH sensitive mesoporous silica-alginate nanoparticles.

The manuscript is interesting and can be accepted for publications after major corrections

1-     First of all the manuscript should be checked by an English native speaker to remove the syntax and typos.

2-     The abstract should be modified to give more digital results rather than elastic sentences.

3-     The first sentence in the introduction is too long, please consider paraphrase

4-     The introduction should provide more details about the previous work

5-     In the introduction “And mesoporous silica nanoparticles MCM-41 is one of the emerging drug loading materials” The sentences should not start with the word “and”

6-     In our experiment, amino modified for MCM- 41 was employed copolycondensation, that is, during the synthesis of mesoporous silicon particles, TEOS is directly added with silane coupling agent, and hydrolytic polycondensation occurs under the protection of template. Pore entrance modification was mainly carried out based on the dense organic layer at the entrance of MCM-41 after modification of organic functional groups in a short time. This method is also convenient for the subsequent realization of amino modified MCM-41, through the protonation of amino under  different pH conditions to achieve the purpose of drug release.”  please consider rephrase 

7-     The novelty of the work should be clearly stated; in addition the authors did not mention any previous work in the literature regarding mesoporous silica nanoparticles.

8-     Please consider using the same font type and size throughout the manuscript

9-     Standard deviations should be added to all figures and tables

10-  The authors were just writing the results with no obvious discussion, the discussion should be improved by correlating results with previously published data, giving an explanation for the obtained results.

11-  Only 13 references are listed in the introduction, no references in the methods or the results are provided   

12-  The authors should use symbols as *, + on all figures to show the statistical differences

13-  Conclusions should be written as a paragraph not in the form of points.

Author Response

1. The discovered syntax errors and typos have been deleted

2. The results have been provided.

3. The first sentence of the introduction has been deleted.

4. The introduction has been fully explained.

It has been amended not to begin with the word "and"

5. Rewriting has been completed

6. This has been supplemented in the conclusions

7. It has been adjusted. The text is in small four font and the text at the bottom of the chart is no. 5

8. Standard deviation has been modified and added

9. Detailed discussions have been made

10. Literature resources of experimental methods have been added

11. Some data have been adjusted

13 The conclusion has been written in a paragraph

Round 2

Reviewer 2 Report

Dear Respected Editor 

The manuscript can be accepted for publication after minor changes

1-     The abstract should be modified to give more digital results rather than elastic sentences.

2-     Standard deviations should be added to table 2

Author Response

1. The abstract section has been modified in detail, and the relevant data results have been listed.

2. Standard deviation data has been added to table 2.